# Peripheral Serotonin Deficiency Affects Anxiety-like Behavior and the Molecular Response to an Acute Challenge in Rats

**DOI:** 10.3390/ijms23094941

**Published:** 2022-04-29

**Authors:** Giulia Sbrini, Sabrina I. Hanswijk, Paola Brivio, Anthonieke Middelman, Michael Bader, Fabio Fumagalli, Natalia Alenina, Judith R. Homberg, Francesca Calabrese

**Affiliations:** 1Department of Pharmacological and Biomolecular Sciences, Università degli Studi di Milano, 20133 Milan, Italy; giulia.sbrini@unimi.it (G.S.); paola.brivio@unimi.it (P.B.); fabio.fumagalli@unimi.it (F.F.); 2Department of Cognitive Neuroscience, Donders Institute for Brain, Cognition and Behavior, Radboud University Nijmegen Medical Centre, 6525 EN Nijmegen, The Netherlands; sabrina.hanswijk@radboudumc.nl (S.I.H.); anthonieke.middelman@radboudumc.nl (A.M.); judith.homberg@radboudumc.nl (J.R.H.); 3Max-Delbrück-Center for Molecular Medicine (MDC), 13125 Berlin, Germany; mbader@mdc-berlin.de (M.B.); alenina@mdc-berlin.de (N.A.); 4Charite-University Medicine, 10117 Berlin, Germany; 5DZHK (German Center for Cardiovascular Research), Partner Site Berlin, 10115 Berlin, Germany; 6Institute for Biology, University of Lübeck, 23562 Lubeck, Germany

**Keywords:** serotonin, tryptophan hydroxylase, acute stress, restraint stress, Bdnf, TPH1

## Abstract

Serotonin is synthetized through the action of tryptophan hydroxylase (TPH) enzymes. While the TPH2 isoform is responsible for the production of serotonin in the brain, TPH1 is expressed in peripheral organs. Interestingly, despite its peripheral localization, alterations of the gene coding for TPH1 have been related to stress sensitivity and an increased susceptibility for psychiatric pathologies. On these bases, we took advantage of newly generated TPH1^−/−^ rats, and we evaluated the impact of the lack of peripheral serotonin on the behavior and expression of brain plasticity-related genes under basal conditions and in response to stress. At a behavioral level, TPH1^−/−^ rats displayed reduced anxiety-like behavior. Moreover, we found that neuronal activation, quantified by the expression of *Bdnf* and the immediate early gene *Arc* and transcription of glucocorticoid responsive genes after 1 h of acute restraint stress, was blunted in TPH1^−/−^ rats in comparison to TPH1^+/+^ animals. Overall, we provided evidence for the influence of peripheral serotonin levels in modulating brain functions under basal and dynamic situations.

## 1. Introduction

In humans, about 95% of whole-body serotonin is localized in the peripheral compartments and is involved in the control of smooth muscle contraction, blood pressure, hemostasis, insulin secretion, and energy metabolism [1,2,3,4,5].

Serotonin is synthetized from the essential amino acid, tryptophan (Trp). In the periphery, the first, rate-limiting step of its synthesis is driven by tryptophan hydroxylase 1, (TPH1) which is selectively expressed in the enterochromaffin cells, in the thymus, in the spleen and in the pineal gland [6,7,8,9,10]. 

Despite the peripheral localization of this enzyme, some single-nucleotide polymorphisms, both in the coding and in the non-coding regions of the human *TPH1* gene, have been associated with an increased susceptibility to depression, post-traumatic stress disorders, and alcohol abuse [11,12,13]. Moreover, low levels of monoamines have been found in the peripheral blood of people suffering from depression [14,15]. However, the mechanisms through which peripheral serotonin could influence brain functionality are not clear, also considering the inability of the serotonin to pass through the blood–brain barrier during adulthood [16]. 

Mice with a genetic deletion of *Tph1* were bred by several laboratories. These mice were instrumental to unraveling the role of peripheral serotonin in platelet function, liver regeneration, the regulation of metabolic processes and adipocyte differentiation, lactation, erythropoiesis, heart development, and inflammatory and fibrotic diseases of gut, pancreas, lung, and liver (reviewed in [17]). However, the impact of *Tph1* deletion on brain function was addressed only in very few studies. Moreover, slight alterations in gait dynamics [18] and no changes in basic behavior were reported in these mice [7,19], whereas the molecular aspects of brain function were not investigated so far.

This study aimed to clarify if the lack of peripheral serotonin affected animal behavior and gene expression in the central nervous system. Although mice were a preferred model in preclinical research during previous decades because of the availability of tools to manipulate their genomes, these tools are now also available for rats [20]. Since rats exhibit a more extensive behavioral repertoire compared to mice, the availability of transgenic rats enriches our arsenal of experimental models in (neuro)science research [20]. Therefore, in this study we employed TPH1-deficient (TPH1^−/−^) rats newly generated at the Gene Editing Rat Resource Center of the Medical College of Wisconsin. As a first step, we measured serotonin and tryptophan levels in different peripheral organs to confirm the lack of TPH1 activity. Next, we exposed TPH1^+/+^ and TPH1^−/−^ rats to the elevated plus maze test to evaluate their anxiety-like behavior, a common feature of a wide range of different psychopathologies linked to *TPH1* polymorphisms [21]. Finally, considering the relevance of the interaction between the genetic background and environment in the development of psychiatric symptoms [22,23,24,25,26], we evaluated if and how TPH1^−/−^ rats responded to an acute challenge by exposing these animals to a single session of acute restraint stress (RS). At the molecular level, we focused on the expression of the neurotrophin brain-derived neurotrophic factors (*Bdnf)* and other immediate early genes, such as *Arc* and *Fos*, as well as the expression of different glucocorticoid responsive genes that were involved in anxiety-phenotype and responded to stress exposure [27,28,29,30].

Molecular analyses aimed to dissect the possible mechanisms that are physiologically activated under baseline conditions and in response to stress [25,26,30] and may be impaired in TPH1^−/−^ rats. Gene expression measurements were conducted in the prefrontal cortex (PFC), a brain region that is sensitive to environmental changes [24,25,30] and known to interact with the hypothalamus–pituitary–adrenal (HPA) axis [31,32,33] and the ventral hippocampus (VHip), another region implicated in this stress response [34]. We discovered that peripheral serotonin contributed to the modulation of brain function in rats under basal and stressful conditions.

## 2. Results

### 2.1. Serotonin Levels Are Dramatically Reduced in Peripheral Organs of TPH1^−/−^ Rats

First, we analyzed serotonin levels in TPH1^−/−^ and TPH1^+/+^ rats. As expected, and in line with data previously obtained in TPH1^−/−^ mice [6,20,35,36], serotonin was almost undetectable in the serotonin-producing tissues of TPH1^−/−^ rats (duodenum: −99% *p* < 0.01 vs. TPH1^+/+^; ileum: −99% *p* < 0.01 vs. TPH1^+/+^; colon: −99% *p* < 0.01 vs. TPH1^+/+^; pineal gland: −99% *p* < 0.01 vs. TPH1^+/+^; Student’s *t*-test) (Figure 1A–D). Although a decrease was observed in blood and spleen, about 5 to 10% of wild-type serotonin levels were still detectable in these tissues (blood: −94% *p* < 0.01 vs. TPH1^+/+^ spleen: −91% *p* < 0.001 vs. TPH1^+/+^, Student’s *t*-test), confirming previous TPH1^−/−^ mouse data (26, 27) (Figure 1E,F). We also evaluated serotonin levels in the PFC and brainstem. As shown in Figure 1G,H, no differences between genotypes were observed in these brain regions, confirming the preservation of the brain serotonin system in the absence of peripheral serotonin, similar to previous data obtained in mice [37,38].

### 2.2. Tryptophan Levels Are Modulated in the Pineal Gland of TPH1^−/−^ Rats

While serotonin concentration was homogenously reduced in peripheral organs, we found that the levels of its precursor tryptophan were increased by almost three times specifically in the pineal glands (+183% *p* < 0.01 vs. TPH1^+/+^; Student’s *t*-test) (Figure 2A). Conversely, we did not find any alteration in Trp levels in the blood, spleen, different gut compartments and brain parts (Figure 2B–H).

### 2.3. TPH1^−/−^ Rats Show Reduced Anxiety in the Elevated Plus Maze Test

Anxiety is a common feature of different psychopathologies that have been linked to human TPH1 polymorphisms [21]. Hence, here we analyzed the anxiety levels of TPH1^+/+^ and TPH1^−/−^ rats by exposing the animals to the elevated plus maze test.

Interestingly, the lack of peripheral serotonin induced a reduction in anxiety behavior. TPH1^−/−^ rats spent more time in the open arms (+63% vs. TPH1^+/+^ *p* < 0.05; Student’s *t*-test; Figure 3A) and in the center of the maze (+33% *p* < 0.05 vs. TPH1^+/+^ *p* < 0.05; Student’s *t*-test; Figure 3B), while they spent less time in the closed arms (−20% *p* < 0.01 vs. TPH1^+/+^ *p* < 0.01; Student’s *t*-test; Figure 3C). Moreover, as shown in Figure 3D, the rats showed a reduced latency to the first entry in the open arms (−41% vs. TPH1^+/+^ *p* < 0.05; Student’s *t*-test).

### 2.4. Baseline Bdnf mRNA Levels Are Increased and Induction of Its Expression Is Impaired after RS in TPH1^−/−^ Rats Specifically in the Prefrontal Cortex

To evaluate if the change in anxiety is paralleled by changes in brain plasticity in TPH1^−/−^ rats, we measured the levels of *Bdnf* in the PFC and in the VHip, both at basal level and after one single session of RS.

We found a significant genotype X RS interaction for total *Bdnf* mRNA levels (F_(1–21)_ = 4.806 *p* < 0.05; two-way ANOVA). In line with this, we found an upregulation of the total neurotrophin expression at basal level in TPH1^−/−^ rats compared to TPH1^+/+^ animals (+56% *p* < 0.05 vs. TPH1^+/+^/No RS; Fisher’s PLSD). However, total *Bdnf* mRNA levels were increased after RS in TPH1^+/+^ rats (+52% *p* < 0.05 vs. TPH1^+/+^/No RS; Fisher’s PLSD) but not in TPH1^−/−^ rats (Figure 4A).

On the contrary, as shown in Figure 4B, we did not find changes in the expression of total *Bdnf* in the VHip, neither due to the genotype nor due to RS, suggesting a specific effect of both factors in the PFC.

### 2.5. The Changes in Total Bdnf Expression in the PFC Are Sustained by Bdnf Long 3′UTR, Bdnf Isoform IV and VI

To further dissect the effect observed for *Bdnf* transcription, we measured the expression of *Bdnf* long 3′UTR and of the two most abundant isoforms in the brain, the isoforms IV and VI, in the PFC.

As shown in Figure 5A,B, we found a similar expression pattern for *Bdnf* long 3′UTR and *Bdnf* isoform IV (*Bdnf* long 3′UTR: genotype X RS interaction: F_(1–19)_ = 5.817 *p* < 0.05; *Bdnf* isoform IV: genotype X RS interaction: F_(1–19)_ = 5.345 *p* < 0.05; two-way ANOVA). Indeed, the lack of peripheral serotonin induced an increase in their mRNA levels in TPH1^−/−^ animals (*Bdnf* long 3′UTR: +35% *p* < 0.05; *Bdnf* isoform IV: +32% *p* < 0.01 vs. TPH1^+/+^/No RS; Fisher’s PLSD), and one hour of RS upregulated the pool of the long transcripts and the isoform IV mRNA levels in wild-type rats (*Bdnf* long 3′UTR: +33% *p* < 0.05; *Bdnf* isoform IV: +43% *p* < 0.05 vs. TPH1^+/+^/No RS; Fisher’s PLSD), while we did not find any further increase for TPH1^−/−^ RS rats.

Finally, we observed a genotype X RS interaction for *Bdnf* isoform VI (F_(1–20)_ = 5.165 *p* < 0.05; two-way ANOVA) with an upregulation of its expression in TPH1^−/−^ No RS animals (+42% *p* < 0.05 vs. TPH1^+/+^/No RS; Fisher’s PLSD), which was reduced after RS (−27% *p* < 0.05 vs. TPH1^+/+^/No RS; Fisher’s PLSD) (Figure 5C).

### 2.6. The Enhancement in the Immediate Early Gene Arc mRNA Levels Due to RS Exposure Is Blunted in TPH1^−/−^ Rats

Given that the response to RS in TPH1^−/−^ rats was blunted in terms of *Bdnf* gene expression, we further evaluated the expression of the two immediate early genes *Arc* and *Fos* that were normally upregulated after acute stimuli [26,30].

Interestingly, we found an effect of RS (F_(1–22)_ = 4.963 *p* < 0.05; two-way ANOVA) and a genotype X RS interaction (F_(1–22)_ = 4.518 *p* < 0.05; two-way ANOVA) on *Arc* expression. Indeed, as shown in Figure 6A, we found an upregulation of its mRNA levels specifically in TPH1^+/+^/RS rats (+57% *p* < 0.01 vs. TPH1^+/+^/No RS; Fisher’s PLSD) with no changes in the TPH1-deficient counterpart.

On the contrary, *Fos* expression was neither affected by RS nor by the genotype (Figure 6B).

### 2.7. Upregulation of Glucocorticoid Responsive Genes Expression Induced by RS Exposure Is Blunted in TPH1^−/−^ Rats

Glucocorticoids play a crucial role in stress response, inducing transcriptional activation of glucocorticoid-dependent genes [25]. Recently, we evaluated the expression pattern of several glucocorticoid-dependent genes in the PFC in response to acute stress [25]. Hence, we chose highly reactive genes for evaluation after RS in the PFC of the TPH1^−/−^ rats.

We found an effect of RS on *Dusp1* expression (F_(1–21)_ = 7.478 *p* < 0.05; two-way ANOVA). Indeed, RS upregulated *Dusp1* expression specifically in TPH1^+/+^ rats (+27% *p* < 0.05 vs. TPH1^+/+^/No RS: Fisher’s PLSD) but not in TPH1^−/−^ rats (Figure 7A).

Similarly, we found an effect of genotype and of RS on *Fkbp5* mRNA levels (genotype: F_(1–22)_ = 6.241 *p* < 0.05; RS: F_(1–22)_ = 5.874 *p* <0.05 two-way ANOVA) and a significant genotype X RS interaction on *Nr4a1* expression (F_(1–23)_ = 6.569 *p* < 0.05; two-way ANOVA). Indeed, we found a specific upregulation of both *Fkbp5* and *Nr4a1* after RS in TPH1^+/+^ rats (*Fkbp5*: +35% *p* < 0.05; *Nr4a1*: +30% *p* < 0.05 vs. TPH1^+/+^/No RS; Fisher’s PLSD) but no changes in TPH1^−/−^ rats (Figure 7B,C).

Moreover, *Sgk1* expression was modulated by RS (F_(1–23)_ = 76.873 *p* < 0.001 two-way ANOVA) with an upregulation of its expression both in TPH1^+/+^ (+95% *p* < 0.001 vs. TPH1^+/+^/No RS; Fisher’s PLSD) and TPH1^−/−^ rats (+71% *p* < 0.001 vs. TPH1^−/−^/No RS; Fisher’s PLSD). However, this increase was more robust in TPH1^+/+^ than in TPH1^−/−^ RS rats (+18% *p* < 0.05 vs. TPH1^+/+^/RS; Fisher’s PLSD) (Figure 7D).

Finally, as shown in Figure 7E,F, we observed an effect of RS on *Gadd45β* and *Gilz* expression (*Gadd45β*: F_(1–23)_ = 33.405 *p* < 0.001; *Gilz*: F_(1–23)_ = 86.627 *p* < 0.001; two-way ANOVA) with a similar upregulation of their expression in TPH1^+/+^ rats (*Gadd45β*: +60% *p* < 0.001; *Gilz*: +103% *p* < 0.001 vs. TPH1^+/+^/No RS; Fisher’s PLSD) and in TPH1^−/−^ rats (*Gadd45β*: +65% *p* < 0.001; *Gilz*: +71% *p* < 0.001 vs. TPH1^−/−^/No RS; Fisher’s PLSD).

## 3. Discussion

In this study, we confirmed our initial hypothesis that peripheral serotonin can modulate brain functions. Specifically, we found that *Tph1*-deletion in rats led to a dramatic decrease in peripheral, but not central, serotonin in these animals. Furthermore, this life-long depletion of peripheral serotonin in rats affected anxiety-like behavior and expression of genes involved in brain plasticity and stress response under basal conditions and after the exposure to RS.

Among the animal models that have been developed to study the role of peripheral serotonin, TPH1-deficient mice exhibit low levels of serotonin in peripheral organs, while, in line with the selective distribution of the TPH1 isoform in the periphery, the levels of serotonin in the central nervous system remain unchanged [6,19,38]. Accordingly, TPH1-deficient mice are a useful tool for studying the role of peripheral serotonin in cardiac function, metabolism, and hemostasis [3,35,36,38,39]. However, for cognitive and molecular neuroscience, rats are the preferred model [20]. Here, for the first time, we characterized TPH1-deficient rats and investigated if a lack of TPH1 had an impact on behavior and brain plasticity.

In line with previous mouse data [7,19,37,38], TPH1-deficient rats showed a reduction in serotonin levels in the periphery. Indeed, the deletion of the *Tph1* gene in rats resulted in a dramatic decrease in serotonin levels, which were almost undetectable in *Tph1*-expressing organs, such as pineal glands and intestinal tissues. In mammals, besides sites of serotonin synthesis, this monoamine is also present in high amounts in platelets, which take it up via serotonin transporter (SERT). Serotonin levels in blood and spleen, the main platelet-containing tissues, were also drastically reduced in TPH1^−/−^ rats, but still accounted for 5–10% of the wild type. The same phenotype was previously observed in TPH1-deficient mice, arguing for the existence of a conserved mechanism to maintain serotonin homeostasis in the absence of TPH1 enzyme, probably through phenylalanine hydroxylase [37].

Similar to TPH1-deficient mice [37], we did not find any alterations in serotonin levels in the PFC and brain stem of TPH1^−/−^ rats, suggesting that the central serotonergic system is rather unaffected by the lack of peripheral serotonin. Hence, the observed central phenotypes must be mediated by other, indirect mechanisms.

Interestingly, while levels of the serotonin precursor Trp were not altered in peripheral tissues, such as blood, spleen, and gut, we found an increase in Trp in the pineal gland. Trp is an essential amino acid, and most dietary Trp is metabolized through the kynurenine pathway, whereas only a small portion (around 1–2%) is utilized for the serotonin production (reviewed in [40]). In the pineal gland, melatonin synthesis from Trp with serotonin as an intermediate product is a main metabolic pathway of Trp. Evidently, the kynurenine pathway [41] is not able to remove the excessive Trp from the pineals in the absence of TPH1, leading to its accumulation in this organ.

Since several studies revealed a link between single-nucleotide polymorphisms in the human *TPH1* gene and an increased susceptibility to developing psychiatric conditions, including depression and anxiety-related disorders [11,12,13,42], we evaluated the anxiety-like behavior of TPH1-deficient rats by exposing them to the elevated plus maze test. The results from this behavioral test indicate that TPH1^−/−^ rats are less anxious when compared to TPH1^+/+^ rats. Indeed, they spent more time in the open arms and in the center of the apparatus, while spending less time in the closed arms, where rats are supposed to feel safer. Moreover, they showed a reduced latency to the first entry in the open arms. Although it has been shown that the lack of peripheral serotonin is able to alter locomotor activity in mice [18], no study to date has evaluated whether the altered function of TPH1 could modulate other behaviors, including those related to psychiatric pathologies. Our study suggests that the lack of peripheral serotonin in TPH1^−/−^ rats has a positive impact on anxiety-like behavior at basal level. Here, we showed that the levels of *Bdnf* are increased under basal conditions in the PFC of TPH1^−/−^ rats, suggesting an interplay between serotonin and *Bdnf* expression in modulating the behavioral phenotype observed in TPH1^−/−^ rats. Yet, the mechanistic links between low serotonin in the periphery, *Bdnf* expression and anxiety-related behavior are to be identified.

In the context of psychiatric illnesses, the genetic background often acts as a risk factor and different molecular alterations can be unmasked only after an environmental stimulus. For example, in our previous studies, we showed that the lack of central serotonin impaired the activation of different systems only after an acute stress, but not under basal conditions [25,26]. Hence, here we exposed TPH1^+/+^ and TPH1^−/−^ rats to a single session of RS to evaluate whether reduced peripheral serotonin levels could interfere with the activation of stress response. We firstly focused on neurotrophic mechanisms and found that the total *Bdnf* mRNA levels were upregulated at basal level in TPH1^−/−^ rats and that this increase was sustained by the expression of the pool containing long transcripts of the neurotrophin and by *Bdnf* isoforms IV and VI. However, while, as expected, TPH1^+/+^ RS rats showed an activation of the system with an increase in total *Bdnf* mRNA levels, *Bdnf* long 3′UTR and *Bdnf* isoform IV, we did not find any further increase in rats lacking peripheral serotonin. Thus, we could hypothesize that peripheral serotonin was able to interfere with the activation of the brain *Bdnf* machinery after RS exposure. In line with these results, by measuring the expression of the immediate early genes *Arc* and *Fos,* which are implicated in acute stress responses [43,44], we found an upregulation of *Arc* specifically in TPH1^+/+^ rats exposed to RS, while a lack of serotonin in peripheral compartments blunted its activation.

To further support this evidence, we also evaluated the transcriptional activation induced by glucocorticoid receptors, which is normally boosted after acute stress exposure due to the activation of the HPA axis and subsequent release of corticosterone [45,46]. We found that the expressions of glucocorticoid-responsive genes—*Dusp1*, *Fkbp5,* and *Nr4a1*—were upregulated specifically in wild-type rats exposed to RS while their transcription remained unchanged in knockout animals. Moreover, *Sgk1* was upregulated in both genotypes, but this increase was higher in TPH1^+/+^ rats. Finally, *Gadd45β* and *Gilz* were equally upregulated after RS both in TPH1^+/+^ and TPH1^−/−^ rats. Although other analyses should confirm this hypothesis, we can speculate here that the absence of peripheral serotonin could interfere with corticosterone release, consequently inhibiting the whole downstream pathway. In line with this, the adrenal gland, from which corticosterone is released, expresses serotonin receptors, and their agonism stimulates the release of this hormone [47]. Moreover, serotonin can stimulate the release of the adrenocorticotropic hormone (ACTH) from the pituitary gland, a central region lacking the blood–brain barrier and exposed to peripheral serotonin [48,49]. Unfortunately, we were not able to measure corticosterone and ACTH release to confirm these hypotheses, which is a limitation of our study. Moreover, not all glucocorticoid-responsive genes show the same activation pattern in TPH1^−/−^ rats, which could be due to other stress-related factors involved in the activation of their transcription [50,51].

While the results obtained in this work argue for a link between peripheral serotonin levels and brain functionality under resting conditions and in response to environmental stimuli, the underlying mechanisms and the direct causality between the lack of peripheral serotonin and brain functions have yet to be fully understood. In particular, ontogenetic analyses at different developmental time points could clarify if the crosstalk between peripheral serotonin and brain wiring starts to occur during development. Indeed, the blood–brain barrier represents a clear boundary at adulthood, but during the first stages of life it is still immature and different molecules, including serotonin, can cross it [52]. Therefore, the reduction in serotonin amounts reaching the brain during the first period of life could interfere with the correct formation of neuronal circuits and inhibit their correct functioning [53]. Moreover, the analysis of the gut microbiome and the gut–brain axis represents another important factor to be studied in TPH1-deficient animals [54]. The evaluation of these aspects may uncover mechanisms linking the lack of peripheral serotonin and behavior and brain plasticity alterations observed in TPH1^−/−^ rats in this study.

## 4. Materials and Methods

### 4.1. Animals

TPH1^−/−^ rats were obtained at the Gene Editing Rat Resource Center of The Medical College Of Wisconsin by CRISPR/Cas9 technology. A Cas9 protein was injected together with a guide RNA, targeting the sequence AGATGTCATTCAGCTGTTCTCGG in the TPH1 gene, into zygotes of Wistar Kyoto rats, producing one base pair deletion in the exon 4 and thereby a frameshift mutation.

The breeding was conducted at the Central Animal Laboratory of the Radboud University Medical Center in Nijmegen, the Netherlands. From birth, the animals were housed under standardized conditions involving a 12/12 h light/dark cycle, at 22 °C and around 80% of humidity with access to food and water ad libitum.

Heterozygous females were bred with heterozygous males to generate TPH1^+/+^ and TPH1^−/−^ experimental animals. At adulthood, male TPH1^+/+^ (*n* = 21) and TPH1^−/−^ (*n* = 35) rats were tested for their anxiety level during the elevated plus maze test. In particular, a plus-shaped platform was located at 50 cm from the floor. The maze consisted of two closed arms (50 cm × 10 cm with 40 cm high walls) and two open arms of the same dimensions without walls. The two equal arms were positioned opposite each other, and the four arms formed a central platform at their intersection. At the beginning of the test, a rat was placed on the central platform looking at one of the open arms. The position and the movements of the animals were recorded for 5 min, and the time spent in the closed arms, in the center, and in the open arms of the arena was analyzed using EthoVision XT (version 3.1., Noldus, Wageningen, The Netherlands) software.

Two weeks later, part of the animals (*n* = 5/6) were exposed to one hour of RS, as previously described in [26], to test their stress responsiveness. Briefly, the animals were placed in an air-accessible apparatus with a similar size to the rats for one hour. Immediately after RS, animals were anesthetized and sacrificed through decapitation. No RS animals (*n* = 5/6) were left undisturbed in their home cage until the sacrifice.

For the molecular analyses, the PFC was dissected according to the atlas of Paxinos and Watson (Paxinos and Watson, 2007) from 2 mm thick slices (plates 6–9, including Cg1, Cg3, and IL sub-regions), whereas vHip were dissected from the whole brain (plates 34–43 of the Paxinos and Watson atlas). The brain regions were immediately frozen on dry ice and stored at −80 °C.

All experimental procedures were approved by the Central Committee on Animal Experiments (Centrale Commissie Dierproeven, CCD, The Hague, The Netherlands), limiting the number of animals used and minimizing animal suffering.

### 4.2. HPLC Analyses

For blood analyses, three hundred microliters of whole blood was collected into 1 mL syringes prefilled with 100 μL of heparin and quickly transferred to Eppendorf tubes containing 10 μL of perchloric acid (PCA) and 5 μL of 10 mg/mL of ascorbic acid, vortexed, and centrifuged (20,000× *g*, 30 min, 4 °C). The supernatant was collected and frozen at −80 °C until high-performance liquid chromatography (HPLC) analyses. For the organ collection, rats were transcardially perfused with ice-cold PBS to remove blood, containing platelet 5-HT. Tissues were snap-frozen in liquid nitrogen and kept at −80 °C. Frozen tissue samples were homogenized in 710 μM ascorbic acid and 2.4% perchloric acid (Sigma-Aldrich, Steinheim, Germany), precipitated proteins were pelleted through centrifugation (20 min, 20,000× *g*, 4 °C), and the collected supernatant was analyzed for serotonergic metabolites (Trp, and 5-HT) using HPLC with fluorometric detection. Samples were separated on a C18 reversed-phase column (Synergi Fusion RP, Phenomenex, Aschaffenburg, Germany) at 20 °C in a 10 mM potassium phosphate buffer (pH 5.0) (Sigma-Aldrich, Steinheim, Germany), with 5% methanol (Roth, Karlsruhe, Germany) and a flow rate of 0.8–1.0 mL/min on a Nexera X2 HPLC system (Shimadzu, Tokyo, Japan). The excitation wavelength was 295 nm, and the fluorescent signal was measured at 345 nm. LabSolutions 5.85 software (Shimadzu, Tokyo, Japan) was used to analyze the peak parameters of chromatographic spectra and quantify substance levels, based on comparative calculations with alternately measured external standards. Amounts of 5-HT and Trp were normalized to the wet tissue weight.

### 4.3. mRNA Extraction and Gene Expression Analyses

Total RNA was isolated by a single step of guanidinium isothiocyanate/phenol extraction using PureZol RNA isolation reagent (Bio-Rad Laboratories, Segrate (Mi), Italy) according to the manufacturer’s instructions and quantified by spectrophotometric analysis. The samples were then processed for a real-time polymerase chain reaction (RT-PCR) to assess the expression of the target genes (primer and probes sequences are listed in Table 1 and Table 2). RNA was treated with DNase (Thermoscientific, Milano, Italy) to avoid DNA contamination. Gene expression was analyzed by TaqMan qRT-PCR one-step RT-PCR kit for probes (Bio-Rad Laboratories, Italy). Samples were run in 384-well formats in triplicate as a multiplexed reaction with a normalizing internal control (36b4). Thermal cycling was initiated with an incubation at 50 °C for 10 min (RNA retro-transcription) and then at 95 °C for 5 min (TaqMan polymerase activation). After this initial step, 39 cycles of PCR were performed. Each PCR cycle consisted of heating the samples at 95 °C for 10 s to enable the melting process and then for 30 s at 60 °C for the annealing and extension reactions. A comparative cycle threshold (Ct) method was used to calculate the relative target gene expression.

### 4.4. Statistical Analyses

All the analyses were conducted by using IBM SPSS Statistics, version 24 or with Microsoft Excel. Data were collected in individual animals and are presented as means ± standard error (SEM).

Results were analyzed with two-way analysis of variance (ANOVA) followed by the Fisher’s protected least significant difference (PLSD) test or with a Student’s *t*-test. Significance for all tests was assumed for *p* < 0.05.

## 5. Conclusions

Overall, these data suggest that, even though serotonin itself cannot cross the blood–brain barrier, the lack of serotonin in the periphery can indeed affect brain functions. This is reflected by reduced anxiety, an upregulation of genes related to neuroplasticity and the glucocorticoid system and altered responsivity of these systems to RS exposure. These findings aid in the understanding of the genetic associations found between *TPH1* gene polymorphisms and stress-related disorders.

## Figures and Tables

**Figure 1 ijms-23-04941-f001:**
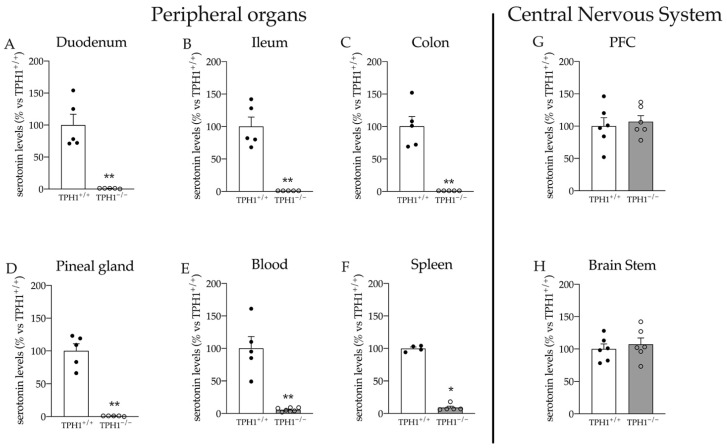
Serotonin levels in the duodenum (**A**), ileum (**B**), colon (**C**), pineal gland (**D**), blood (**E**), spleen (**F**), PFC (**G**) and in the brain stem (**H**) of TPH1^+/+^ and TPH1^−/−^ rats. Data are presented as means ± standard error (SEM) of at least 4 independent determinations. * *p* < 0.05 ** *p* < 0.01 vs. TPH1^+/+^; Student’s *t*-test.

**Figure 2 ijms-23-04941-f002:**
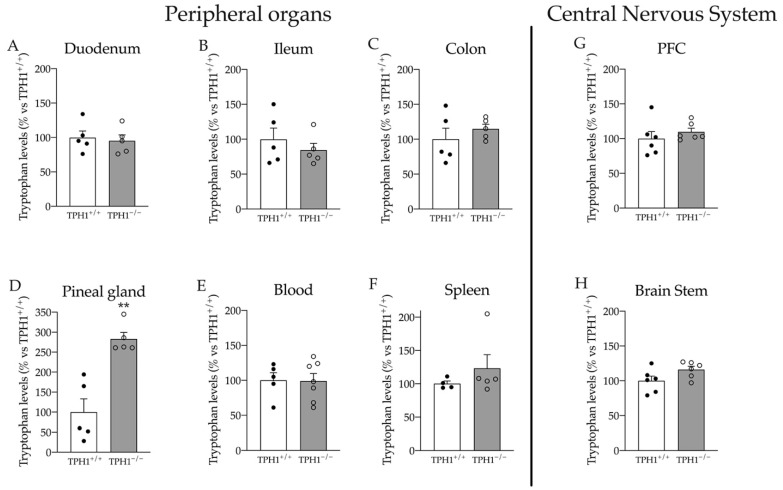
Tryptophan levels in the duodenum (**A**), ileum (**B**), colon (**C**), pineal gland (**D**), blood (**E**), spleen (**F**), PFC (**G**) and in the brain stem (**H**) of TPH1^+/+^ and TPH1^−/−^ rats. Data are presented as means ± standard error (SEM) of at least 4 independent determinations. ** *p* < 0.01 vs. TPH1^+/+^; Student’s *t*-test.

**Figure 3 ijms-23-04941-f003:**
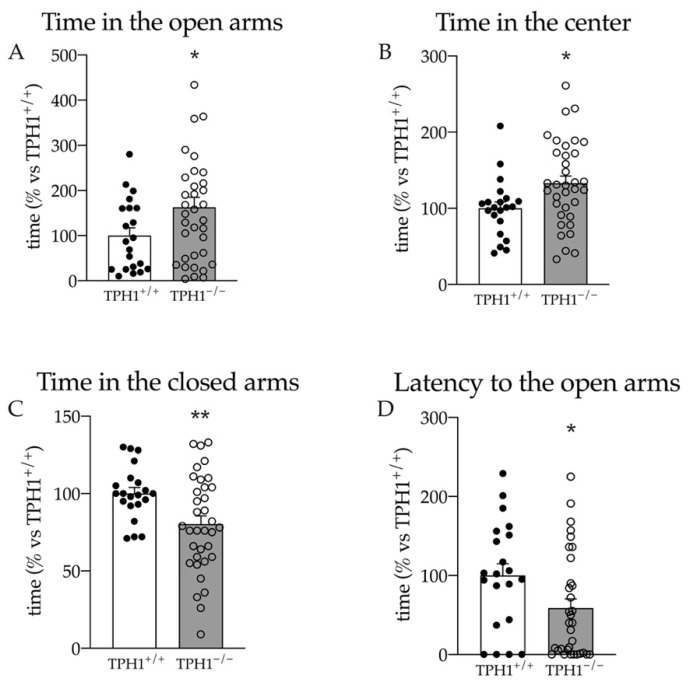
Elevated plus maze test results. Time spent in the open arms (**A**), in the center (**B**), in the closed arms (**C**) and latency to the first entrance in the open arms (**D**). Data are presented as mean ± standard error of the mean (SEM) of at least 21 independent determinations. * *p* < 0.05; ** *p* < 0.01 vs. TPH1^+/+^; Student’s *t*-test.

**Figure 4 ijms-23-04941-f004:**
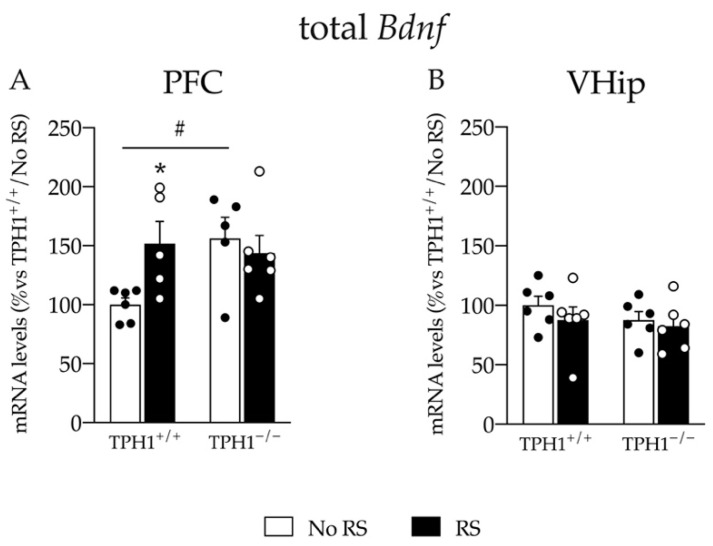
Analyses of total *Bdnf* mRNA levels in the PFC (**A**) and in the VHip (**B**) of TPH1^+/+^ and TPH1^−/−^ male adult rats exposed to RS and sacrificed immediately after stress. The data, expressed as percentage of TPH1^+/+^/No RS (set at 100%), are the mean ± standard error of the mean (SEM) of at least 5 independent determinations. * *p* < 0.05 vs. naïve for the same genotype; # *p* < 0.05, vs. TPH1^+/+^ for the same condition, two-way ANOVA with Fisher’s PLSD.

**Figure 5 ijms-23-04941-f005:**
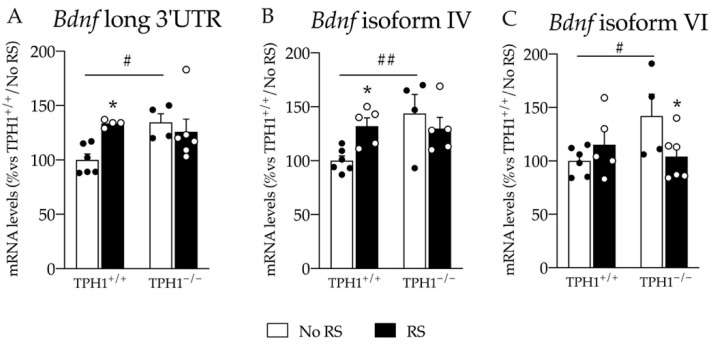
Analyses of *Bdnf* long 3′UTR (**A**), *Bdnf* isoforms IV (**B**) and VI (**C**) mRNA levels in the PFC of TPH1^+/+^ and TPH1^−/−^ male adult rats exposed to RS and sacrificed immediately after stress. The data, expressed as percentage of TPH1^+/+^/No RS (set at 100%), are the mean ± standard error of the mean (SEM) of at least 4 independent determinations. * *p* < 0.05 vs. naïve for the same genotype; # *p* < 0.05, ## *p* < 0.01 vs. TPH1^+/+^ for the same condition, two-way ANOVA with Fisher’s PLSD.

**Figure 6 ijms-23-04941-f006:**
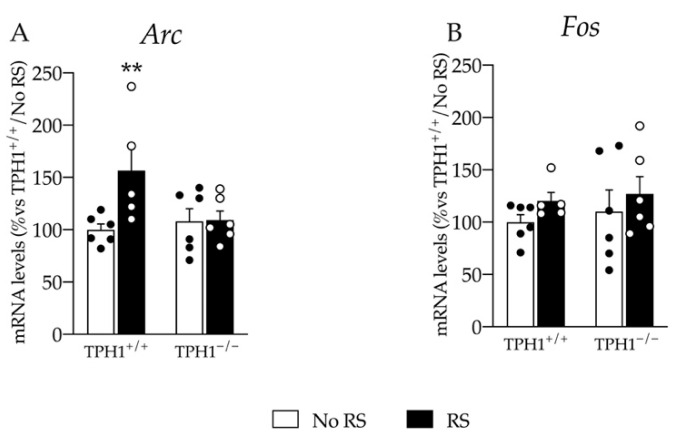
Analyses of *Arc* (**A**) and *Fos* (**B**) mRNA levels in the PFC of TPH1^+/+^ and TPH1^−/−^ male adult rats exposed to RS and sacrificed immediately after stress. The data, expressed as a percentage of TPH1^+/+^/No RS (set at 100%), are the mean ± standard error of the mean (SEM) of at least 5 independent determinations. ** *p* < 0.01 vs. naïve for the same genotype; two-way ANOVA with Fisher’s PLSD.

**Figure 7 ijms-23-04941-f007:**
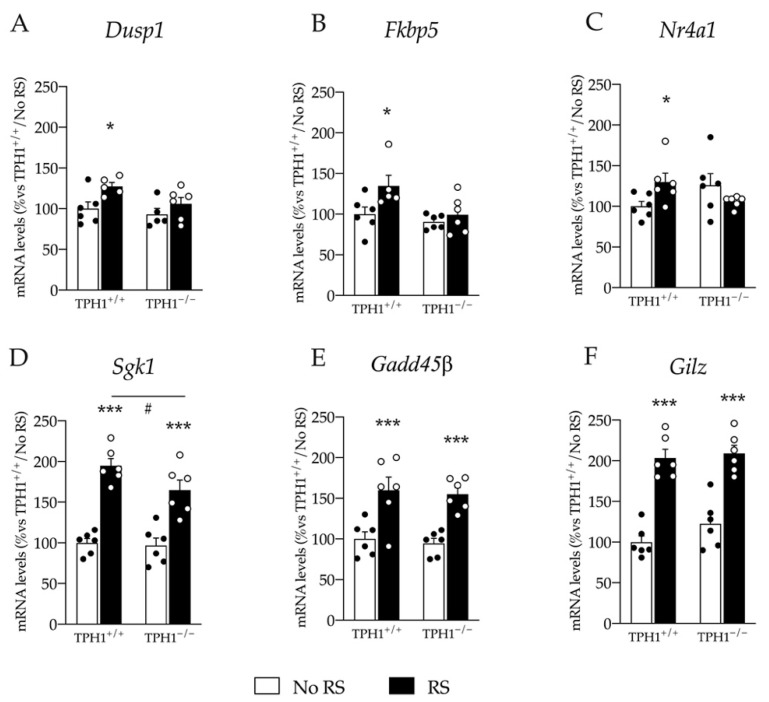
Analyses of *Dusp1* (**A**), *Fkbp5* (**B**), *Nr4a1* (**C**), *Sgk1* (**D**), *Gadd45β* (**E**) and *Gilz* (**F**) mRNA levels in the PFC of TPH1^+/+^ and TPH1^−/−^ male adult rats exposed to RS and sacrificed immediately after stress. The data, expressed as percentage of TPH1^+/+^/No RS (set at 100%), are the mean ± standard error of the mean (SEM) of at least 5 independent determinations. * *p* < 0.05; *** *p* < 0.001 vs. naïve for the same genotype; # *p* < 0.05 vs. TPH1^+/+^ for the same condition; two-way ANOVA with Fisher’s PLSD.

**Table 1 ijms-23-04941-t001:** Primers and probes sequences acquired from Eurofins MWG-Operon.

Gene	Forward Sequence	Reverse Sequence	Probe Sequence
Total *Bdnf*	AAGTCTGCATTACATTCCTCGA	GTTTTCTGAAAGAGGGACAGTTTAT	TGTGGTTTGTTGCCGTTGCCAAG
*Arc*	GGTGGGTGGCTCTGAAGAAT	ACTCCACCCAGTTCTTCACC	GATCCAGAACCACATGAATGGG
*Fos*	TCCTTACGGACTCCCCAC	CTCCGTTTCTCTTCCTCTTCAG	TGCTCTACTTTGCCCCTTCTGCC
*Dusp1*	TGTGCCTGACAGTGCAGAAT	ATCTTTCCGGGAAGCATGGT	ATCCTGTCCTTCCTGTACCT
*Fkbp5*	GAACCCAATGCTGAGCTTATG	ATGTACTTGCCTCCCTTGAAG	TGTCCATCTCCCAGGATTCTTTGGC
*Sgk1*	GACTACATTAATGGCGGAGAGC	AGGGAGTGCAGATAACCCAAG	TGCTCGCTTCTACGCAGC
*Gilz*	CGGTCTATCAACTGCACAATTTC	CTTCACTAGATCCATGGCCTG	AACGGAAACCACATCCCCTCCAA
*36b4*	TCAGTGCCTCACTCCATCAT	AGGAAGGCCTTGACCTTTTC	TGGATACAAAAGGGTCCTGG

**Table 2 ijms-23-04941-t002:** Primers and probes sequences acquired from Life Technologies.

Gene	Accession Number	Assay ID
*Bdnf* long 3′UTR	EF125675.1	Rn02531967_s1
*Bdnf* isoform IV	EF125679.1	Rn01484927_m1
*Bdnf* isoform VI	EF125680.1	Rn01484928_m1
*Nr4a1*	BC097313.1	Rn01452530_g1
*Gadd45β*	BC085337.1	Rn01452530_g1

## Data Availability

The data presented in this study are available on request from the corresponding author.

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
