# Peer review of "Peripheral Serotonin Deficiency Affects Anxiety-like Behavior and the Molecular Response to an Acute Challenge in Rats"

_ijms, 2022, doi:10.3390/ijms23094941_

Round 1

Reviewer 1 Report

Remarks to the Author:

The manuscript “Peripheral serotonin deficiency alters brain homeostasis and the response to an acute challenge in rats” reports attempt to characterize alterations in behavior and in expression of brain plasticity related genes caused by chronical absence of the peripheral serotonin. For this purpose, the authors measure anxiety level with the elevated plus maze test and RNA levels of several genes with qRT-PCR in the prefrontal cortex of TPH1-/- rats under basal conditions and in response to acute restrain stress. While there is always room for new phenotypic observations in science, the incongruities between the findings reported in the manuscripts and the conclusions, lack of hypothesis and proper priors are disturbing.

Major limitation:

Authors confirm that in TPH1-/- rats serotonin level in periphery organs drops similarly as it has been shown previously in TPH1-/- mice. Authors acknowledge that depletion is not full and around 5% of normal serotonin level is still present in blood and spleen. In the discussion authors reference to Mordhorst et al 2021, where phenylalanine hydroxylase (PAH) was identified as a possible alternative to TPH1 for peripheral serotonin synthesis. Peripheral serotonin has been shown to play important roles in platelets; inflammatory and fibrotic diseases of gut, pancreas, lung, and liver; in the regulation of lactation, heart development, adipocyte differentiation, and erythropoiesis; and in liver regeneration. It is unclear which of those processes are impaired in TPH1−/− rats and how much PAH ameliorate absence of TPH1. In discussion section authors speculated that serotonin level might control corticosterone release and this might be the path for peripheral serotonin to influence the central nervous system processes. The gain and loss of function approaches need to be performed to establish causality between peripheral serotonin, corticosterone release and brain functions.

To interpret behavior and expression results presented by the authors it is essential to know what is serotonin level in the brain of TPH1-/- rats especially in dorsal and median raphe nucleus as well as in the prefrontal cortex. Also explain why expression analysis were performed in the prefrontal cortex, any neuronal circuits level motivations or may be where is a special expression pattern of serotonin or glucocorticoid or mineralocorticoid receptors?

Authors confirm previously reported results (see Mosienko et al. Transl Psychiatry 2012, https://doi.org/10.1038/tp.2012.44) that animals lacking peripheral serotonin display reduced anxiety-like behavior in the elevated plus maze. This might be not a direct effect, the rescue experiment is missing.   

Authors use several terms - “Naïve stress”, “acute stress”, “dynamic situations” and “stress” - for the acute restraint (RS) experiment. To define what type of stress animals experience confirm serum corticosterone (CORT) and/or epinephrine (EPI) levels after RS, use consistence terminology throughout the manuscript.

In previous work (Front. Cell. Neurosci. 2020 https://doi.org/10.3389/fncel.2020.00128) authors reported changes in expression of several genes in the prefrontal cortex of TPH2-deficient (TPH2−/−) rats. Some of the genes overlap with the results presented in current manuscript. Authors didn't provide sufficient rational for the genes chosen for the study, the discussion of expression results is controversial and have not been integrated with authors previous work and existing literature.

Overall the hypothesis, discussion and the citation of the literature in the text are confusing.

Minor comments:

Both groups in Figure 4 C and D labels are indicated as the TPH1+/+ rats

Add individual data points to all bar charts.

Author Response

Please see the file attached

Reviewer 2 Report

This is a well written paper, however it is impossible to determine its value due to the lack of n values. Currently no mention of the number of animals included in each group exists. Further it should be mentioned if TPH+/+ are age matched or siblings or run in parallel to determine the potential impact of variance in the study. This need to be added before the study can be evaluated in full.

Author Response

Please see the file attached

Reviewer 3 Report

This manuscript showed that peripheral serotonin deficiency alters brain homeostasis and the response to an acute challenge in rats. Interestingly, the lack of serotonin in the periphery can indeed affect brain functions.

There are some following questions worthy of discussion:

  1. Please deleted the dot after the title.
  2. Did you detect Trp and 5-HT in the blank blood samples? I just wonder whether there is interference in the blank sample. Please provide the HPLC chromatogram of the blank blood samples.
  3. Line 318: Please change “over” to “on”.
  4. Line 335: Please change “384 well” to “384-well”.
  5. Line 347: Please change “means±standard error” to “means ± standard error”.
  6. As for the references, please formulate a unified format, especially for their titles.
  7. What’s the meaning of “naïve” in the manuscript?
  8. Description in Conflicts of Interest section is too simply and not clear.
  9. The resolution of all the figures need to be improved.

Author Response

Please see the file attached

Round 2

Reviewer 1 Report

Authors did a great job revising the manuscript by addressing most of concerns, improving results presentation and interpretation.

Major comments:

Please adjust the title of paper to better match the scope and main findings.

Although Authors agreed in their reply that a rescue experiment is interesting, they have not performed any related experiments and took the major limitation of their study as a suggestion for the future experiments. There is a significant body of the literature since 1950s on the peripheral serotonin and available techniques on how to properly manipulate 5-HT levels.

Restore at least acutely the physiological serotonin levels in TPH1−/− rats and report effects on anxiety-like behavior as well as total Bdnf and Arc mRNA levels in the PFC of naïve rats and rats exposed to RS.

Minor:

Now the aims of the study stated: “This study aimed to clarify if the lack of peripheral serotonin affects brain functionality.” Change “brain functionality” to “animal behavior and expression of several genes in central nervous system”.

Line 63 change “neuroplasticity genes” to “immediate early genes”

Author Response

Please see the file attached

Reviewer 3 Report

No more suggestions.

Author Response

Thank you